# PIG-1 MELK-dependent phosphorylation of nonmuscle myosin II promotes apoptosis through CES-1 Snail partitioning

Hai Wei[1¤], Eric J. Lambie[1,2], Daniel S. Osório[3], Ana X. Carvalho[3], Barbara Conradt[1,2,4]*

**1** Department Biology II, Faculty of Biology, Ludwig-Maximilians-University Munich, Großhadener, Planegg-Martinsried, Germany, **2** Research Department of Cell and Developmental Biology, Division of Biosciences, University College London, Gower Street, London WC1E 6BT, United Kingdom, **3** Instituto de Investigação e Inovação em Saúde, Universidade do Porto, Portugal, **4** CIPSM–Center for Integrated Protein Science Munich, Butenandtstraße, München, Germany

¤ Current address: UT Southwestern Medical Center, Department of Pharmacology, Dallas Texas, United States of America
* b.conradt@ucl.ac.uk

**Editor:** shai shaham, the rockefeller university, UNITED STATES MINOR OUTLYING ISLANDS

**Data Availability Statement:** All relevant data are within the manuscript and its Supporting Information files.

## Abstract

The mechanism(s) through which mammalian kinase MELK promotes tumorigenesis is not understood. We find that the *C. elegans* orthologue of MELK, PIG-1, promotes apoptosis by partitioning an anti-apoptotic factor. The *C. elegans* NSM neuroblast divides to produce a larger cell that differentiates into a neuron and a smaller cell that dies. We find that in this context, PIG-1 MELK is required for partitioning of CES-1 Snail, a transcriptional repressor of the pro-apoptotic gene *egl-1* BH3-only. *pig-1* MELK is controlled by both a *ces-1* Snail- and *par-4* LKB1-dependent pathway, and may act through phosphorylation and cortical enrichment of nonmuscle myosin II prior to neuroblast division. We propose that *pig-1* MELK-induced local contractility of the actomyosin network plays a conserved role in the acquisition of the apoptotic fate. Our work also uncovers an auto-regulatory loop through which *ces-1* Snail controls its own activity through the formation of a gradient of CES-1 Snail protein.

## Author summary

Apoptosis is critical for the elimination of 'unwanted' cells. What distinguishes wanted from unwanted cells in developing animals is poorly understood. We report that in the *C. elegans* NSM neuroblast lineage, the level of CES-1, a Snail-family member and transcriptional repressor of the pro-apoptotic gene *egl-1*, contributes to this process. In addition, we demonstrate that *C. elegans* PIG-1, the orthologue of mammalian proto-oncoprotein MELK, plays a critical role in controlling CES-1 Snail levels. Specifically, during NSM neuroblast division, PIG-1 MELK controls partitioning of CES-1 Snail into one but not the other daughter cell thereby promoting the making of one wanted and one unwanted cell. Furthermore, we present evidence that PIG-1 MELK acts prior to NSM neuroblast division by locally activating the actomyosin network.

**Funding:** This work was supported by LMU Munich (https://www.uni-muenchen.de/index.html), the Deutsche Forschungsgemeinschaft (Center for Integrated Protein Science Munich – CIPSM, DFG EXC 114 to B.C., https://www.dfg.de/en/) and the European Research Council (https://erc.europa.eu/) under the European Union's Horizon 2020 research and innovation programme (grant agreement 640553 – ACTOMYO to A.X.C.). H.W. was supported by a predoctoral fellowship from the China Scholarship Council (https://www.csc.edu.cn/). A.X.C. has a Principal Investigator position from the Fundação para a Ciência e Tecnologia (https://www.fct.pt/) (CEECIND/01967/2017). The funders had no role in study design, data collection and analysis, decision to publish, or preparation of the manuscript.

**Competing interests:** The authors have declared that no competing interests exist.

## Introduction

Mammalian 'Maternal Embryonic Leucine zipper Kinase' referred to as MELK was identified in 1997 using two different approaches. Using differential cDNA display, transcripts of the MELK gene were identified in early mouse embryos and shown to be maternally expressed [1]. Transcripts of the same gene (initially referred to as MPK38) were also identified using a PCR-based screen for new kinase genes expressed in a murine teratocarcinoma cell line [2]. The MELK gene encodes a member of the family of AMPK (AMP-activated protein kinase) -related serine/threonine protein kinases and has orthologues in species as diverse as *C. elegans* and humans [3]. Vertebrate MELK kinase has been implicated in a broad range of cellular processes such as cell division, cell death and survival, cellular differentiation and embryonic development, and many of the proposed functions appear to be context-dependent. Therefore, it has been suggested that MELK may affect various aspects of cell fate acquisition [3–5]. This is supported by the finding that MELK can affect gene expression through physical interactions with transcription factors and regulators of protein synthesis [6–8]. Furthermore, through its interaction with the scaffold protein Arrestin 3, MELK protein may act in signaling cascades that are restricted to certain subcellular regions [5]. The MELK gene is overexpressed in different types of cancers, including 'triple negative' breast cancer (TNBC), the most aggressive form of this type [9]. Importantly, it was recently demonstrated that MELK function is required for clonogenic growth of TNBC-derived cells, and this has made MELK a target for the development of novel cancer therapeutics [10–13]. However, the physiological function(s) of MELK and the mechanism(s) through which its overexpression contributes to tumorigenesis remain largely unknown.

The *C. elegans* orthologue of the MELK gene, *pig-1*, was identified in several independent genetic screens and subsequent analyses have implicated *pig-1* in asymmetric cell division and programmed cell elimination. Originally, the *pig-1* gene was identified in a screen for mutations that alter the numbers of two specific types of neurons, the HSN and PHB neurons, and was subsequently found to affect the asymmetric division of the HSN/PHB neuroblast in larvae of the first larval stage (L1 larvae) [14]. The HSN/PHB neuroblast normally divides asymmetrically to give rise to a larger cell that divides to generate one HSN and one PHB neuron and a smaller cell that undergoes apoptosis [15]. In *pig-1* mutants, the HSN/PHB neuroblast divides symmetrically to give rise to two cells of similar sizes, both of which survive and can divide to give rise to a total of two HSNs and two PHBs [14]. Subsequent analyses in early *C. elegans* embryos confirmed a role for *pig-1* in asymmetric cell division and demonstrated that this function is not specific to cell divisions that give rise to an apoptotic death [16–18]. Furthermore, in this context, *pig-1* was found to act in a pathway that is redundant with a pathway that is dependent on the gene *par-1*, which encodes a serine/threonine kinase similar to PIG-1 [16]. Several observations suggest that *pig-1* acts in asymmetric cell division by controlling the distribution of nonmuscle myosin II NMY-2 in dividing cells [18, 19]. However, the mechanism(s) through which *pig-1* MELK affects NMY-2 distribution remains unknown. *pig-1* was furthermore identified in a screen for mutations that prevent the shedding or extrusion of inappropriately surviving 'undead' cells from the embryo proper in mutants that lack the caspase gene *ced-3* and in a screen for mutations that block the *ced-3* caspase-dependent apoptotic death of specific cells during development [20, 21]. Based on this it was proposed that *pig-1* MELK is a component of a 'programmed cell elimination' pathway that acts in parallel to the canonical *ced-3* caspase-dependent apoptosis pathway to ensure that specific cells are reproducibly removed during *C. elegans* development. Finally, mammalian AMPK-related kinases are activated through phosphorylation by the kinase LKB1 at a conserved threonine residue and this is dependent on the LKB1 binding partners STRADα and MO25α [22]. The loss of *C.*

*elegans par-4*, *strd-1* or *mop-25.1*, *2*, which encode *C. elegans* homologs of LKB1, STRADα and MO25α, respectively, phenocopies the loss of *pig-1* in the context of both asymmetric cell division and programmed cell elimination [20, 23], indicating that the activity of PIG-1 MELK could be dependent on PAR-4 LKB1-dependent PIG-1 phosphorylation.

Here we study one specific apoptotic death during *C. elegans* development, the death of the two Neuro-Secretory Motorneuron (NSM) sister cells in the NSM neuroblast lineage. The two NSM neuroblasts (NSMnb) divide to each give rise to a larger cell, the NSM, which differentiates into a serotonergic neuron, and a smaller cell, the NSM sister cell (NSMsc), which dies through apoptosis [15]. The *C. elegans* gene *ces-1* encodes a zinc-finger DNA binding protein and member of the family of Snail-like transcription factors. A dominant mutation in a *cis*-regulatory region of the *ces-1* gene (*n703*) referred to as *ces-1* 'gain-of-function' (gf) mutation causes the overexpression of the gene in the NSMnb and this results in symmetric NSMnb division and the generation of two cells of similar sizes, both of which survive and differentiate into NSM-like neurons [24–26]. We have previously shown that the *pig-1* MELK gene is a direct transcriptional target of CES-1 Snail and that the *ces-1* gf mutation affects asymmetric NSMnb division by repressing *pig-1* expression in the NSMnb [27]. We now demonstrate that *pig-1* activity in the NSM neuroblast lineage is also controlled through the *par-4* LKB1, *strd-1* STRADα and *mop-25.1*, *2* MO25α-dependent pathway. In addition, we present evidence that *pig-1* MELK-dependent phosphorylation of nonmuscle myosin II NMY-2 is required for asymmetric cortical enrichment of NMY-2 in the NSMnb and the generation, during NSMnb division, of differently-sized cells. Furthermore, *pig-1* MELK and *nmy-2* nonmuscle myosin II are also required for the establishment or maintenance of a gradient of CES-1 Snail protein in the NSMnb and the partitioning of CES-1 protein into the larger daughter cell, the NSM. Our results suggest that the resulting lower concentration of CES-1 Snail protein in the NSMsc contributes to the rapid kinetics with which the pro-apoptotic BH3-only gene *egl-1* (the transcription of which is directly repressed by CES-1 Snail) is transcriptionally upregulated and apoptotic cell death is executed in the NSMsc. Hence, our results suggest that in the NSM neuroblast lineage, *pig-1* MELK- and *nmy-2* nonmuscle myosin II-dependent cortical contractility of the actomyosin network is critical not only for daughter cell size asymmetry but the acquisition of the apoptotic fate.

## Results

### *pig-1* MELK may act with *par-4* LKB1, *strd-1* STRADα and *mop-25.1*, *.2* MO25α to control daughter cell size asymmetry in the NSM neuroblast lineage

In wild-type animals, the NSM neuroblast (NSMnb) divides asymmetrically by size to give rise to a smaller cell, the NSM sister cell (NSMsc), and a larger cell, the NSM, with a cell volume ratio of NSMsc to NSM of about 0.64 as measured using a transgene that expresses a mCherry fusion protein that localizes to the plasma membrane and, hence, allows the visualization of cell boundaries ($P_{pie-1}mCherry::ph^{PLC\delta}$) [25] (Fig 1A and 1B). We previously showed that in animals homozygous for the strong *pig-1* loss-of-function (lf) mutation *gm344*, the NSMnb divides symmetrically with a cell volume ratio of NSMsc to NSM of about 1.00 [27] (Fig 1; S1 Fig). It has been shown that in other cellular contexts, *pig-1* MELK acts in a pathway with *par-4* LKB1, *strd-1* STRADα and *mop-25.1*, *.2* MO25α [20, 23]. (Denning et al., 2012 demonstrated that *mop-25.1* and *mop-25.2* are functionally redundant. For this reason, we inactivated the two genes simultaneously.) Using the temperature-sensitive (ts) *par-4* loss-of-function (lf) mutation *it57*ts, the *strd-1* lf mutation *ok2283* and RNA-mediated interference (RNAi) against *mop-25.1* and *mop-25.2* in the background of the *mop-25.2* lf mutation *ok2073* (RNAi[*mop-*

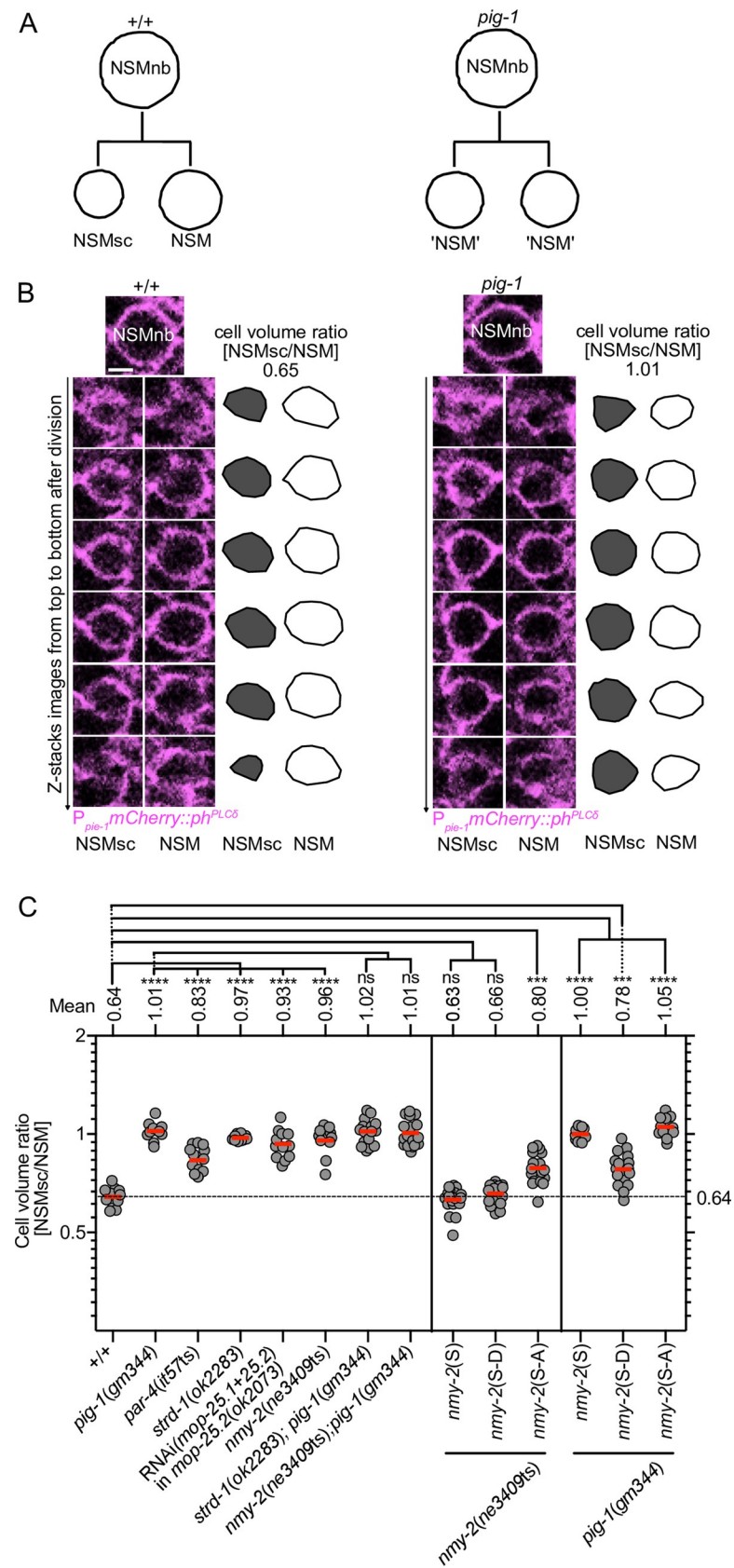

**Fig 1. Control of daughter cell sizes in the NSM neuroblast lineage. (A)** Schematic of the NSMnb division in wild type [+/+] and *pig-1(gm344)* mutant. **(B)** Series (Z-stacks from top to bottom) of fluorescence confocal images of the NSM and NSMsc immediately after NSMnb division in representative wild-type and *pig-1(gm344)* embryo expressing the transgene P$_{pie-1}$*mCherry::ph*$^{PLC\delta}$ (*ltIs44*), which expresses a fusion protein that labels the cell boundary (magenta). Schematic representations of the areas of the NSM (white) and NSMsc (grey) are shown to the right of the images and corresponding cell volume ratios [NSMsc/NSM] are indicated on the top. Scale bar 2 μm. **(C)** Cell volume ratios [NSMsc/NSM] in different genotypes. All strains were homozygous for transgene P$_{pie-1}$*mCherry::ph*$^{PLC\delta}$ (*ltIs44*) except for *par-4(it57*ts), which was homozygous for transgene P$_{pie-1}$*gfp::ph*$^{PLC\delta}$ (*bcIs57*). *nmy-2*(S), *nmy-2*(S-D) and *nmy-2* (S-A) are the integrated single-copy transgenes P$_{nmy-2}$*nmy-2* (*bcSi97*), P$_{nmy-2}$*nmy-2*$^{S211DS1974D}$ (*bcSi102*) and P$_{nmy-2}$*nmy-2*$^{S211AS1974A}$ (*bcSi101*), respectively. Each grey dot represents the ratio of one pair of daughter cells (n = 10–24). Red horizontal lines represent the mean ratios obtained for the different genotypes, which are also indicated on top. The black dotted horizontal line at a ratio of 0.64 represents the mean ratio in wild type. Statistical significance of multiple genotypes compared to a control genotype was determined using the Dunnett's multiple comparisons test (****, P≤0.0001; ns, not significant). Statistical significance between two genotypes was determined using the Mann–Whitney test (***, P≤0.001).

25.1+*mop-25.2*] in *mop-25.2*[*ok2073*]), we found that the loss of *par-4*, *strd-1* or *mop-25.1*, *.2* causes the NSMnb to divide symmetrically by size as well (Fig 1C; S1 Fig). Furthermore, we found that the simultaneous loss of *pig-1* and *strd-1* causes a phenotype that is not different from the phenotypes observed in the single mutants. These results indicate that *par-4* LKB1, *strd-1* STRADα and *mop-25.1*, *.2* MO25α are required for daughter cell size asymmetry in the NSM neuroblast lineage and that in this context, they may act in the same pathway as *pig-1* MELK.

## *pig-1* MELK acts with *par-4* LKB1, *strd-1* STRADα and *mop-25.1*, *.2* MO25α to promote the death of the NSM sister cell

In wild-type animals, the NSMsc dies shortly after being generated and this results in one NSM neuron on each side of the anterior pharynx as shown in larvae of the L3 or L4 stage using a transgene that expresses a nuclear-localized GFP fusion protein in the NSM neuroblast lineage (P$_{tph-1}$*his-24::gfp*) [28] (Fig 2A and 2B). We previously showed that *pig-1(gm344)* causes about 2% of the NSMsc to inappropriately survive with the affected animals exhibiting more than two NSM-like cells in the anterior pharynx (Fig 2B) and that NSMsc survival increases to about 30% in the background of a weak *ced-3* lf mutation, *n2427* [27] (Fig 2C). We now find that *par-4(it57ts)*, *strd-1(ok2283)* or RNAi(*mop-25.1*+*mop-25.2*) in *mop-25.2*(*ok2073*) also causes about 2% NSMsc survival in a wild-type and about 30% NSMsc survival in the *ced-3* (*n2427*) background (Fig 2C). Furthermore, the loss of *strd-1* in *pig-1(gm344)* mutants does not further enhance NSMsc survival in the wild-type or *ced-3(n2427)* background.

In *C. elegans*, cell corpses adopt a refractile morphology, which can be visualized by Nomarski Optics [29]. In wild-type animals, the NSMsc becomes refractile about 22 min after being generated (Fig 3A, 3B and 3C; S2 Fig). We previously showed that in *pig-1(gm344)* mutants, the NSMsc takes about 30 min to become refractile, suggesting a delay in cell death [27] (Fig 3C). We find a similar delay in animals lacking *par-4*, *strd-1* or *mop-25.1*, *.2*. Further-more, in animals lacking both *pig-1* and *strd-1*, this delay is not increased (Fig 3C). Taken together, these results indicate that *par-4* LKB1, *strd-1* STRADα, *mop-25.1*, *.2* MO25α and *pig-1* MELK act in the same genetic pathway to promote the death of the NSMsc.

## Loss of *pig-1* MELK causes delayed transcriptional upregulation of the BH3-only gene *egl-1* in the NSM sister cell

The NSMsc dies through apoptotic cell death [30]. A hallmark of apoptosis during *C. elegans* development is the transcriptional upregulation in cells 'programmed' to die of

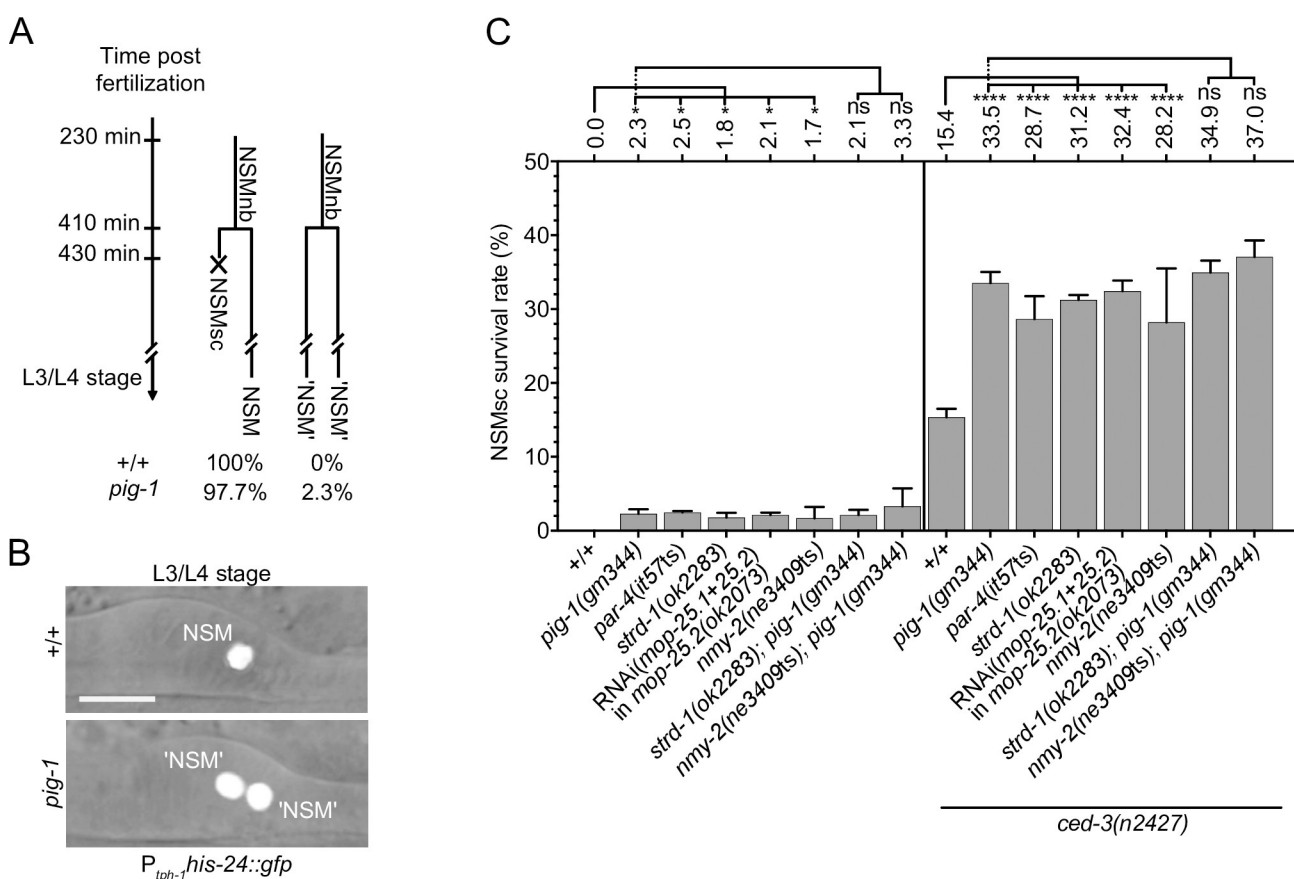

**Fig 2. Control of NSM sister cell death. (A)** Schematic representation of the NSMnb lineage in wild type [+/+] and *pig-1*(*gm344*) mutant. In wild type, the NSMsc dies. In the *pig-1*(*gm344*) lineage shown, the NSMsc inappropriately survives. **(B)** Using the reporter P*tph-1his-24::gfp* (*bcIs66*), the NSM and inappropriately surviving 'undead' NSMsc can be visualized in the anterior pharynx in wild type and *pig-1*(*gm344*) mutants. **(C)** NSMsc survival (%) in different genotypes (n = 200–260). All strains were homozygous for the transgene P*tph-1his-24::gfp* (*bcIs65* in the case of strains containing the mutations *par-4*(*it57*ts), *strd-1*(*ok2283*) or *mop-25.2*(*ok2073*) and *bcIs66* in the case of all other strains). Statistical significance of multiple genotypes compared to a control genotype was determined using the Dunnett's multiple comparisons test (*, P≤0.05, ****, P≤ 0.0001; ns, not significant). Statistical significance between two genotypes was determined using the Mann–Whitney test (ns, not significant).

the gene *egl-1*, which encodes a pro-apoptotic member of the Bcl-2 family of proteins (a BH3-only protein) [31–33]. To determine whether the loss of *pig-1* MELK blocks or delays the death of the NSMsc by compromising the transcriptional upregulation of *egl-1* BH3-only, we used a multi-copy transgene of a transcriptional *egl-1* reporter (P*egl-1his-24::gfp*) [34] to analyze the kinetics of *egl-1* expression in the NSMsc starting immediately after the completion of NSMnb division (0 min post NSMnb division). In wild-type animals, *egl-1* expression becomes detectable about 12 min post NSMnb division at which time point its expression (as measured by fluorescence intensity per area in a focal plane through the center of the NSMsc) is increased 2-fold above baseline (Fig 4A and 4B; S3 Fig). Hence, *egl-1* expression becomes detectable 10 min prior to the death of the NSMsc at 22 min. We found that in *pig-1*(*gm344*) mutants, *egl-1* expression reaches a level that is 2-fold above baseline about 22 min post NSMnb division, which is 12 min prior to the death of the NSMsc at 34 min. Therefore, the loss of *pig-1* MELK causes a delay in the transcriptional upregulation of *egl-1* BH3-only in the NSMsc and this delay correlates with the delay observed in the execution of the NSMsc death.

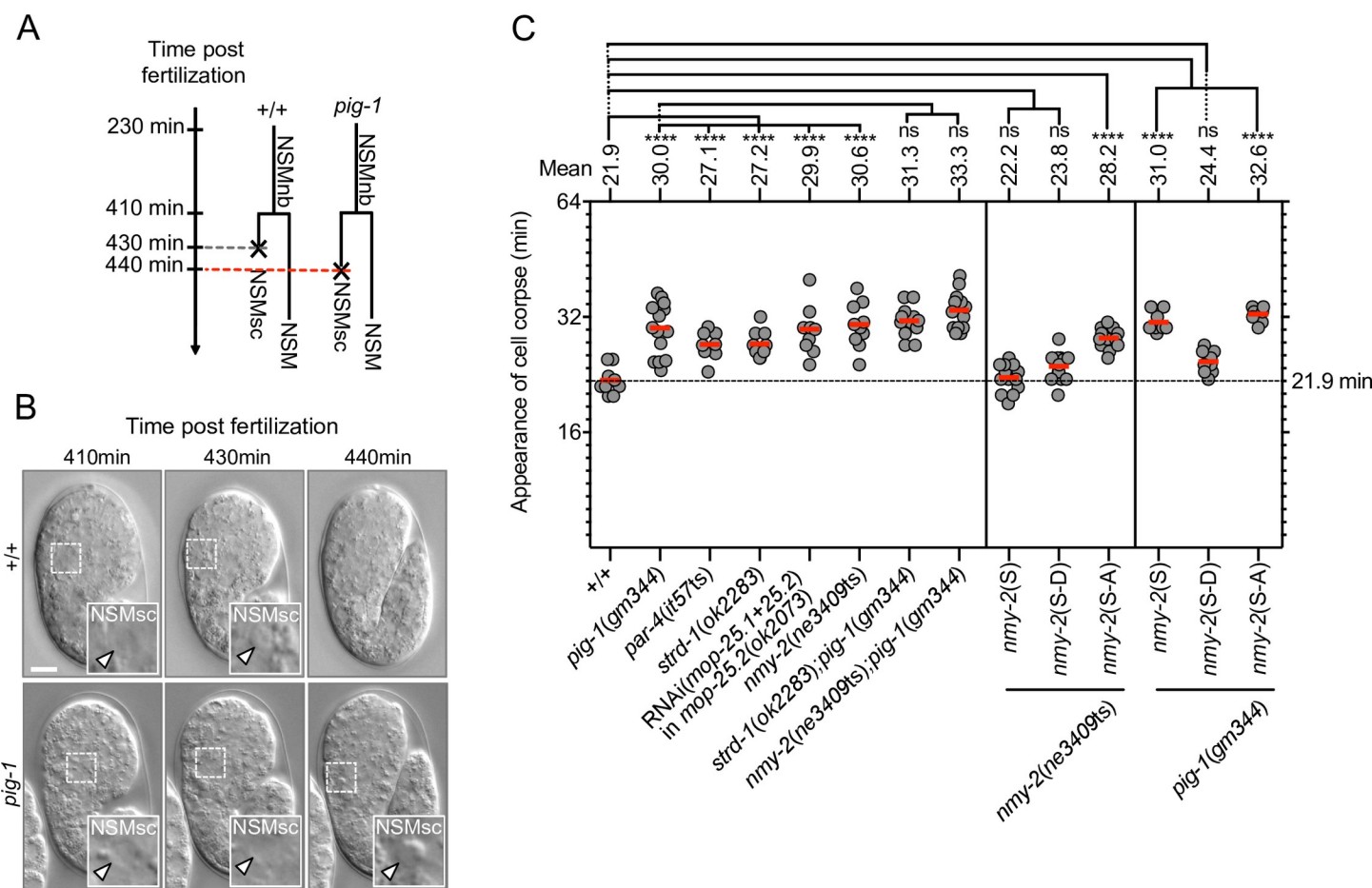

**Fig 3. Kinetics of NSM sister cell death. (A)** Schematic representation of the NSMnb lineage in wild type [+/+] and *pig-1*(*gm344*) mutant. In the *pig-1* lineage shown, the death of the NSMsc is delayed. **(B)** Nomarski images of representative wild-type and *pig-1*(*gm344*) embryo at different times post fertilization [min]. 410 min is immediately post NSMnb cytokinesis. Insets show the NSMsc and white arrow heads point to relevant cells. Scale bar 10 μm. **(C)** Kinetics of NSMsc death as measured by 'Appearance of cell corpse' [min] post NSMnb cytokinesis in different genotypes. *nmy-2*(S), *nmy-2*(S-D) and *nmy-2*(S-A) are the integrated single-copy transgenes $P_{nmy-2}nmy-2$ (*bcSi97*), $P_{nmy-2}nmy-2^{S211DS1974D}$ (*bcSi102*) and $P_{nmy-2}nmy-2^{S211AS1974A}$ (*bcSi101*), respectively. Each dot represents an individual NSMsc (n = 8–14). Red horizontal lines represent the mean times obtained for a given genotype, which is also indicated on top. The black dotted horizontal line at 21.9 min represents the mean in wild type. Statistical significance of multiple genotypes compared to a given control genotype was determined using the Dunnett's multiple comparisons test (****, P≤0.0001; ns, not significance). Statistical significances compared between two genotypes were determined using the Mann–Whitney test (****, P≤0.0001; ns, not significance).

## Formation of a CES-1 Snail gradient in the NSM neuroblast and partitioning of CES-1 Snail into the NSM during NSM neuroblast division

*egl-1* BH3-only transcription in the NSMnb lineage can directly be repressed by the Snail-like transcription factor CES-1 [26, 34]. To visualize CES-1 protein in the NSM neuroblast lineage in real time, we generated a single-copy transgene of a translational *ces-1* reporter that produces a CES-1::mNeonGreen fusion protein under the control of the endogenous *ces-1* promoter ($P_{ces-1}ces-1::mNeonGreen$). This transgene is functional as it rescues a putative *ces-1* null mutation, *tm1036* (S4 Fig). In animals carrying this transgene, CES-1 is detectable in the NSMnb, the NSM and the NSMsc. In the NSMnb, signal is detectable at similar levels throughout the cell about 5 min before it enters metaphase (Fig 5A, +/+, [Before NSMnb division: ~5 min before]). This is confirmed by measuring, in overlapping steps, total fluorescence intensity per area along the dorsal-ventral axis in the representative cell and focal plane shown in Fig 5A

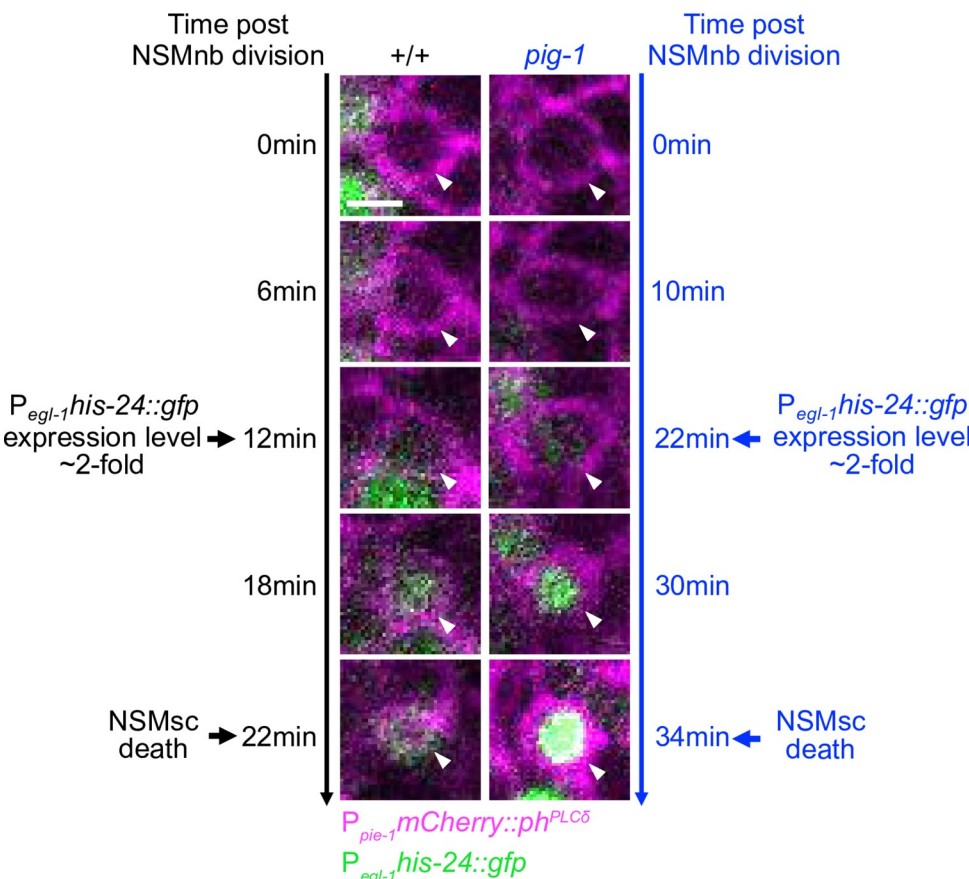

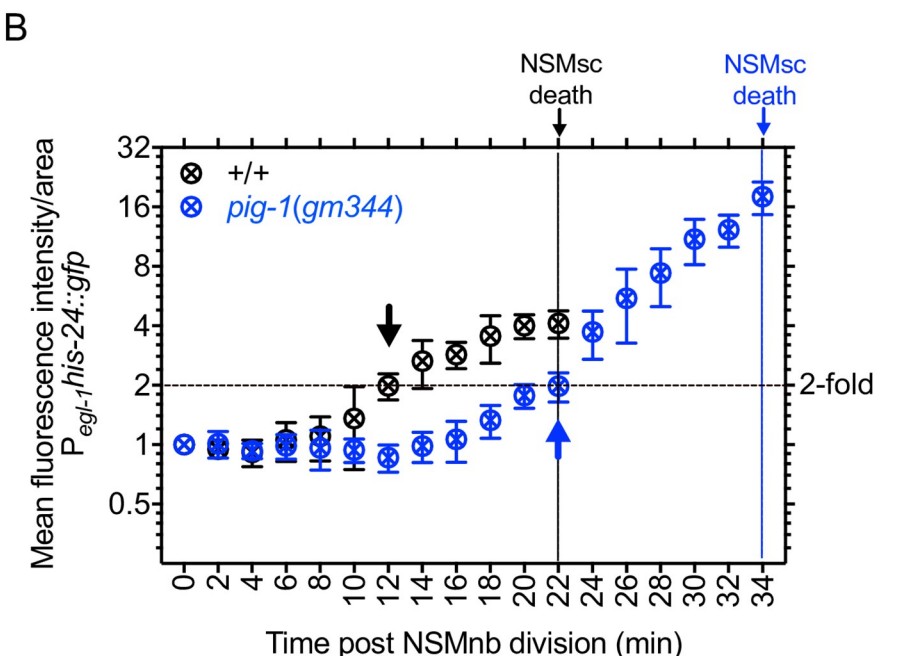

**Fig 4. Kinetics of upregulation of *egl-1* BH3-only expression in the NSM sister cell. (A)** Fluorescence images of NSMsc in representative wild-type [+/+] and *pig-1*(*gm344*) embryo carrying the transgene P$_{egl-1}$*his-24::gfp* (*bcIs37*) (green) at different time points post NSMnb division [min]. Cell boundaries are labeled with transgene P$_{pie-1}$*mCherry::* ph$^{PLC\delta}$ (*ltIs44*) (magenta). Shown are center Z-slices of Z-stacks through the NSMsc. White arrow heads point to NSMsc. Scale bar 2 μm. **(B)** Quantification of P$_{egl-1}$*his-24::gfp* expression. Shown are mean fluorescence intensities per area + SEM in the NSMsc in wild type [+/+] and *pig-1*(*gm344*) mutants at different time points post NSMnb division [min] (n = 3). The data was normalized to the value at time point 0 min, which was set to 1. Black symbols represent wild type and blue symbols represent *pig-1*(*gm344*). Black and blue arrows indicate the time points at which P$_{egl-1}$*his-24::gfp* expression reaches a value of 2 in wild type or *pig-1*(*gm344*), respectively. These time points also represent the time points at which expression becomes detectable by eye. Black dotted horizontal line represents a value of 2. Vertical black line at 22 min indicates the time of NSMsc death in wild type and vertical blue line at 34 min indicates the time of NSMsc death in *pig-1*(*gm344*).

(Fig 5B, +/+). In contrast, once the NSMnb enters metaphase (as indicated by the distinctive rounding up of the cell), a gradient of signal is detectable with increasing signal from the dorsal to the ventral side (Fig 5A, +/+, [Before NSMnb division: metaphase]; Fig 5C, +/+). (The dorsal side will form the NSMsc and the ventral side the NSM.) This gradient results in a ratio of CES-1 in the ventral half of the NSMnb to the dorsal half of 1.45 (Fig 5D, +/+, [Before NSMnb division: Metaphase]). Furthermore, after NSMnb division, less CES-1 is detectable in the NSMsc compared to the NSM (Fig 5A, +/+, [Post NSMnb division: NSMsc, NSM]) with a ratio of CES-1 in the NSM to the NSMsc of 1.51 (Fig 5D, +/+, [Post NSMnb division: NSM/NSMsc]). Based on these results we conclude that prior to NSMnb division, a gradient of CES-1 Snail protein is formed in the NSMnb and that this gradient results in the partitioning of CES-1 Snail predominantly into the NSM.

## *pig-1* MELK but not *ced-1* MEGF10 is required for CES-1 Snail gradient

The bZIP transcription factor CES-2 is orthologous to human DBP (D-box binding PAR bZIP transcription factor) and is thought to directly repress the transcription of the *ces-1* Snail gene in the NSM neuroblast lineage by binding to the *cis*-acting element that is mutated in animals carrying the *ces-1* gf mutation [24, 26, 35]. Indeed, the loss of *ces-2* recapitulates the phenotype of *ces-1* gf animals in the NSM neuroblast lineage [24–27]. Furthermore, using the P$_{ces-1}$*ces-1::* *mNeonGreen* reporter, we observe increased levels of CES-1 Snail protein in the NSM neuroblast lineage of animals lacking *ces-2* function (Fig 5A, *ces-2*(*bc213*)). Importantly, CES-1 protein does not only contribute to the repression of *egl-1* BH3-only transcription in the NSM but also to the polarization and asymmetric division of the NSMnb [25, 27]. For this reason, in animals homozygous for the strong *ces-2* lf mutation *bc213*, increased levels of CES-1 protein in the NSM neuroblast lineage cause the NSMnb to divide symmetrically, resulting in two cells of similar sizes and fates [25]. We find that in this case, there is also no CES-1 gradient detectable in the NSMnb at metaphase and CES-1 is equally segregated into the NSM and NSMsc with a ratio of CES-1 in the NSM to the NSMsc of 0.97 (Fig 5A–5D, *ces-2*(*bc213*)). Furthermore, we previously showed that a low level of CED-3 caspase activity exists in the NSMnb prior to division; this activity is distributed in a gradient that is reciprocal to that of CES-1 Snail protein, i.e., with a higher level of CED-3 caspase activity in the dorsal part of the NSMnb and a lower level in the ventral part [36, 37]. In addition, we demonstrated that the formation and/or maintenance of this gradient is dependent on components of the two *C. elegans* engulfment pathways, for example the gene *ced-1*, which encodes a MEGF10-like scavenger receptor that in this context acts most likely in cells adjacent to the NSMnb [36, 38, 39]. To test whether the CES-1 Snail gradient is also dependent on components of the engulfment pathways, we analyzed animals homozygous for the strong *ced-1* lf mutation *e1735*. We found that the loss of *ced-1* has no effect on the formation of the CES-1 gradient in the NSMnb or the partitioning of CES-1 into the NSM (Fig 5, *ced-1*(*e1735*)).

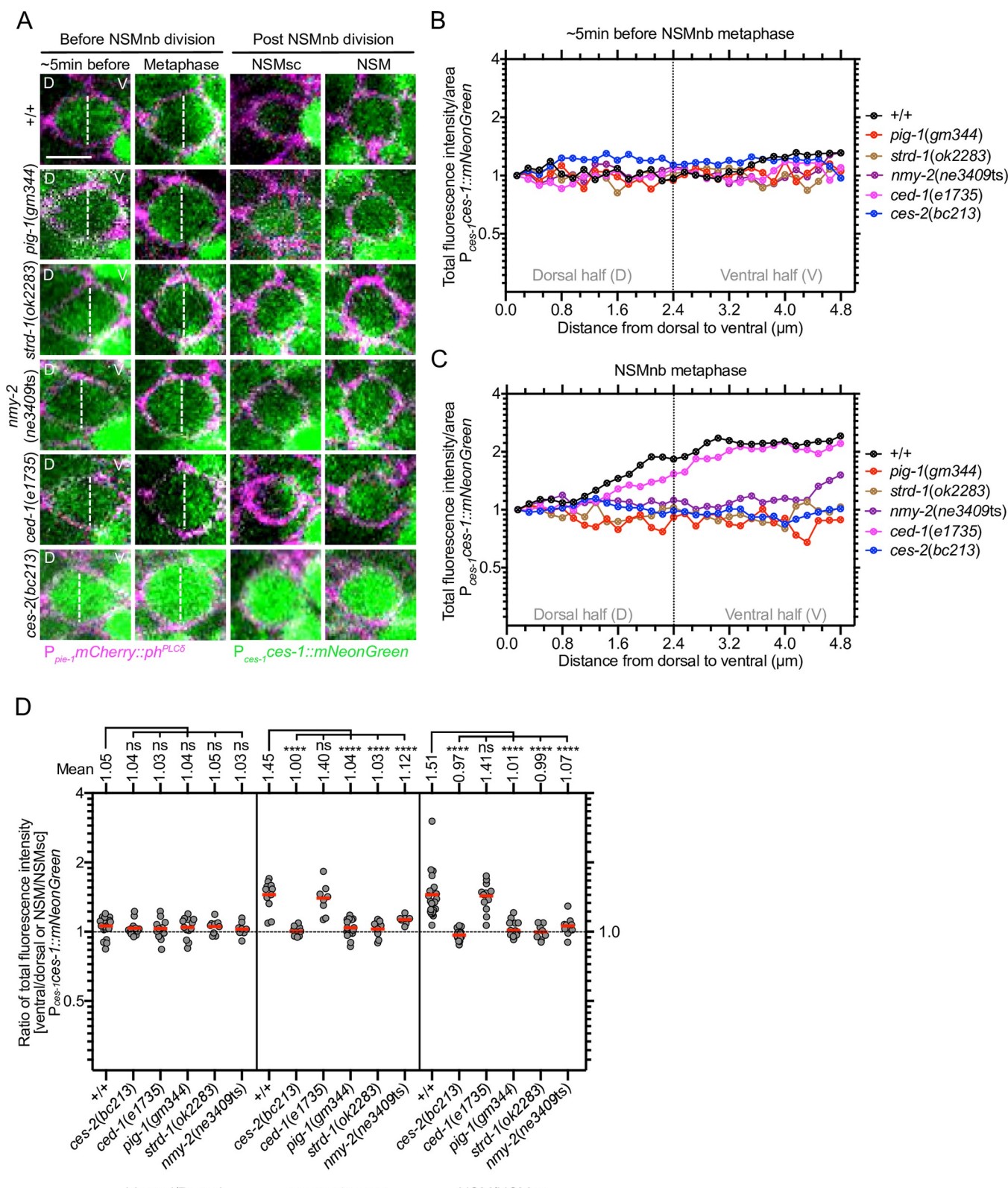

**Fig 5. Formation of a gradient of CES-1 Snail in the NSM neuroblast and CES-1 Snail partitioning during NSM neuroblast division. (A)** Confocal images of representative NSMnb ~5 min before metaphase and at metaphase ('Before NSMnb division'), and NSMnb daughter cells, NSMsc and NSM, immediately after NSMnb division ('Post NSMnb division') in representative NSMnb of wild type [+/+] and various mutants. CES-1 protein is visualized using the transgene P$_{ces-1}$ces-1::mNeonGreen (bcSi66) (green) and cell boundaries are labeled using the transgene P$_{pie-1}$mCherry::ph$^{PLC\delta}$ (ltIs44) (magenta). Shown are maximum intensity projections of Z-stacks for CES-1::mNeonGreen and center Z-slices of Z-stacks for mCherry::PH$^{PLC\delta}$. D and V indicate dorsal and ventral side of NSMnb. White dotted lines indicate middle of NSMnb. Scale bar 2 μm. **(B), (C)** Quantification of CES-1 levels. Shown are total fluorescence intensities per area of CES-1::mNeonGreen from dorsal to ventral side of NSMnb ~5 min before metaphase (B) and at metaphase (C) of the representative cells shown in (A). For each genotype, the values obtained were normalized to the value at distance 0.0 μM, which was set to 1. Black dotted vertical line represents the middle of the NSMnb. **(D)** Ratios of total fluorescence intensities in the ventral to dorsal half of the NSMnb or in the NSM to NSMsc. Each grey dot represents the ventral/dorsal ratio obtained for one NSMnb or the NSMsc/NSM ratio in one pair of daughter cells (n = 7–24). Red horizontal lines represent the means of the ratios obtained for a given genotype and mean values are shown on top. Black dotted horizontal line represents a ratio of 1.0. Statistical significance was determined using the Dunnett's multiple comparisons test (****, P≤0.0001; ns, not significant).

Next, we tested animals lacking *pig-1* MELK or *strd-1* STRADα and found that while levels of CES-1 protein appear unaffected in these animals, no CES-1 gradient is detectable in the NSMnb at metaphase (ratios of CES-1 in the ventral to dorsal side of the NSMnb of 1.04 or 1.05, respectively) and the amount of CES-1 segregated into both daughter cells is similar (ratios of CES-1 in the NSM to NSMsc of 1.01 or 0.99, respectively) (Fig 5, *pig-1*(*gm344*), *strd-1*(*ok2283*)). In summary, these results suggest that the formation of the CES-1 Snail gradient and the partitioning of CES-1 Snail into the NSM require *pig-1* MELK and are under the control of both a *ces-2* DBP, *ces-1* Snail- and *par-4* LKB1, *strd-1* STRADα, *mop-25.1*, *.2* MO25α-dependent pathway, but not a *ced-1* MEGF10-dependent pathway.

## *pig-1* MELK is required for the enrichment of nonmuscle myosin II NMY-2 on the cell cortex of the ventral side of the NSM neuroblast prior to its division

The *C. elegans* nonmuscle myosin II NMY-2 has been proposed to be a target of PIG-1 MELK [19, 40]. Furthermore, NMY-2 phosphorylation at two specific residues, serine 211 (S211) and serine 1974 (S1974), was found to be reduced in *pig-1* mutants [41]. Using a *nmy-2* CRISPR knock-in allele that produces a functional NMY-2::GFP fusion protein (*nmy-2*(*cp13*), *nmy-2*::*gfp+LoxP*) [42], we visualized NMY-2 in the NSMnb. We found that starting about 10 min before NSMnb metaphase, NMY-2 becomes enriched on the cell cortex of the ventral side of the NSMnb and this enrichment is maintained through metaphase (Fig 6A; S5 Fig, +/+). We confirmed this enrichment by measuring, in overlapping steps, fluorescence intensity per area along the dorsal-ventral axis in the representative cell and focal plane shown in Fig 6A (Fig 6B and 6C, +/+). In addition, we measured fluorescence intensities in the dorsal and ventral half of the NSMnb and determined their ratio. This revealed that about 5 min before NSMnb metaphase or at metaphase, there is 1.88-fold or 1.67-fold more NMY-2 in the ventral half of the NSMnb compared to its dorsal half, respectively (Fig 6D, +/+). Since NMY-2 has been proposed to be a target of PIG-1 MELK, next, we analyzed *pig-1*(*gm344*) animals and found that the enrichment of NMY-2 on the cell cortex of the ventral side about 5 min before NSMnb division or at metaphase is almost abolished with ratios of NMY-2 in ventral to dorsal halves of NSMnb of 1.18 or 1.07, respectively (Fig 6A–6D, *pig-1*(*gm344*)). In contrast, the enrichment of NMY-2 at the cleavage furrow is unaffected in *pig-1*(*gm344*) animals (S5A Fig, *pig-1*(*gm344*)). Therefore, nonmuscle myosin II NMY-2 becomes enriched on the cell cortex of the ventral side of the NSMnb prior to its division and this enrichment is dependent on *pig-1* MELK. We also noted a slight, statistically not significant decrease in total fluorescence intensity of NMY-2::GFP in the NSMnb but not the entire embryo, which suggests that the loss of *pig-1* may also affect the turn-over of NMY-2 protein in the NSM neuroblast lineage (S6 Fig).

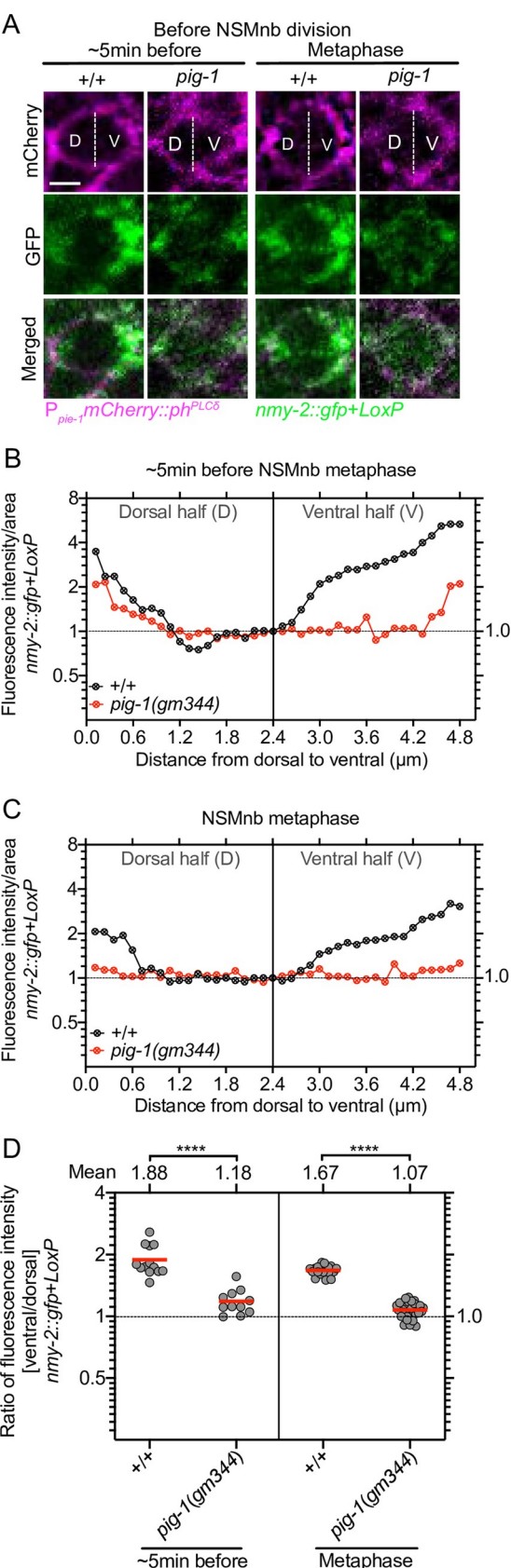

**Fig 6. Asymmetric enrichment of nonmuscle myosin NMY-2 in the NSM neuroblast. (A)** Confocal images of NMY-2::GFP (*nmy-2::gfp+LoxP*, *nmy-2(cp13)*) (green) in representative NSMnb in wild type [+/+] and *pig-1(gm344)* mutants ~5 min before NSMnb metaphase and at NSMnb metaphase. Cell boundaries are labeled with transgene P*_{pie-1}mCherry::ph^{PLCδ}* (*ltIs44*) (magenta). For both transgenes, the center Z-slices of Z-stacks through the NSMnb are shown. D and V indicate dorsal and ventral half of NSMnb. White dotted lines indicate middle of NSMnb. Scale bar 2 μm. **(B), (C)** Shown are fluorescence intensities per area of NMY-2::GFP from dorsal to ventral side of NSMnb ~5 min before metaphase (B) and at metaphase (C) of the representative cells and Z-slices shown in (A). For each genotype, the values obtained were normalized to the value at distance 2.4 μM, which was set to 1. Black dotted horizontal line indicates value 1.0. **(D)** Ratios of fluorescence intensities in the ventral to dorsal half of the NSMnb. Each grey dot represents the ventral/dorsal ratio obtained for one NSMnb (n = 11–28). Red horizontal lines represent the means of the ratios obtained for a given genotype and mean values are shown on top. Black dotted horizontal line represents a ratio of 1.0. Statistical significance was determined using the Mann–Whitney test (****, P≤0.0001).

### *nmy-2* nonmuscle myosin II acts downstream of *pig-1* MELK to control cell size, partitioning of CES-1 Snail into the NSM, and NSM sister cell death

To determine whether *nmy-2* nonmuscle myosin II plays a role in the asymmetric division of the NSMnb, we analyzed animals homozygous for the temperature-sensitive *nmy-2* lf mutation *ne3409*ts. At the non-permissive temperature of 25˚C, *nmy-2(ne3409*ts) causes 100% embryonic lethality (Emb phenotype) (S7 Fig). Therefore, we allowed *nmy-2(ne3409*ts) embryos to develop at the permissive temperature and shifted them to 25˚C 15 min before NSMnb division. We found that the loss of *nmy-2* causes a phenotype that is indistinguishable from that of *pig-1* mutants. Specifically, in *nmy-2(ne3409*ts) animals, the NSMnb divides symmetrically by size (Fig 1C; S1 Fig), 2.1% of the NSMsc survive (Fig 2C), cell death in the NSMsc is delayed (Fig 3C; S2 Fig), and the formation of the CES-1 Snail gradient in the NSMnb and CES-1 partitioning into the NSM are disrupted (Fig 5). Furthermore, the defects in *nmy-2(ne3409*ts) animals in daughter cell sizes (Fig 1C; S1 Fig) and NSMsc death (Fig 2C; Fig 3C; S2 Fig) are not enhanced by the simultaneous loss of *pig-1*. This suggests that in the context of the NSM neuroblast lineage, *pig-1* MELK and *nmy-2* nonmuscle myosin II act in the same genetic pathway.

As mentioned above, phosphorylation of NMY-2 protein at S211 and S1974 is reduced in *pig-1* MELK mutants. To determine whether phosphorylation at these two residues plays a role in the NSM neuroblast lineage, we generated *nmy-2* transgenes that mediate the expression of wild-type NMY-2 protein (referred to as NMY-2(S)) or mutant NMY-2 protein in which S211 and S1974 are replaced with non-phosphorylatable alanines (NMY-2(S-A)) or phospho-mimetic aspartic acids (NMY-2 (S-D)) (S8A Fig). First we analyzed these transgenes for their abilities to rescue the Emb phenotype caused by *nmy-2(ne3409*ts) at 25˚C and found that the *nmy-2*(S) and *nmy-2*(S-D) transgenes rescue the phenotype by 94% or 86%, respectively, whereas the *nmy-2*(S-A) transgene rescues it only by 24% (S7 Fig). This indicates that phosphorylation at S211 and S1974 is critical for the essential function(s) of NMY-2 in embryogenesis.

In embryos rescued from lethality, we then analyzed the NSM neuroblast lineage and found that the defects in daughter cell sizes and cell death in *nmy-2(ne3409*ts) mutants are rescued by the wild-type *nmy-2*(S) and phospho-mimetic *nmy-2*(S-D) transgenes but not the non-phosphorylatable *nmy-2*(S-A) transgene (Fig 1C; Fig 3C). This suggests that phosphorylation of nonmuscle myosin II NMY-2 at these two residues is also critical for the ability of NMY-2 to cause asymmetric NSMnb division and to promote NSMsc death.

Finally, we tested the *nmy-2* transgenes for their abilities to rescue the loss of *pig-1* MELK in the context of the NSM neuroblast lineage. We found that the wild-type *nmy-2*(S) transgene and the non-phosphorylatable *nmy-2*(S-A) transgene fail to rescue the defects in daughter cell size asymmetry and NSMsc death in *pig-1(gm344)* mutants (Fig 1C; Fig 3C). In contrast, the

phospho-mimetic *nmy-2*(S-D) transgene partially rescues the defect in daughter cell sizes (Fig 1C) and fully rescues the defect in NSMsc death (Fig 3C). Based on these findings, we propose that *nmy-2* nonmuscle myosin II acts downstream of *pig-1* MELK to cause asymmetric NSMnb division and that PIG-1 MELK-dependent phosphorylation at S211 and S1974 is critical for NMY-2 function in this context.

## Discussion

### Roles for actomyosin network in NSM neuroblast division

We present evidence in support of the notion that *pig-1* MELK controls daughter cell size asymmetry and CES-1 Snail gradient formation in the NSM neuroblast lineage by directly or indirectly causing the phosphorylation at S211 and S1974 of nonmuscle myosin II NMY-2 (**Fig** 7A). PIG-1 is structurally and functionally similar to the *C. elegans* kinase PAR-1 [16], which has been shown to physically interact with NMY-2 [40]. Therefore, we consider it likely that PIG-1 directly interacts with and phosphorylates NMY-2. Our data also suggest that *pig-1*-dependent phosphorylation of NMY-2 causes or maintains the enrichment of NMY-2 on the cell cortex on the ventral side of the NSMnb that occurs prior to NSMnb division. Whether PIG-1 protein itself is enriched in this part of the NSMnb is currently not known. Asymmetric cortical enrichment of myosin can cause long-range intracellular flows and asymmetric cortical contractility of the actomyosin network [43–46]. Therefore, we propose that the asymmetric cortical enrichment of NMY-2 in the NSMnb results in flows directed towards the ventral side and increased cortical contractility on the ventral side (**Fig** 7B). How intracellular protein gradients, such as the CES-1 Snail gradient described here, are established is not fully understood. Studies of the one-cell *C. elegans* embryo suggest that diffusion-state switching and long-range cortical flows can generate stable protein gradients [47–52]. Furthermore, asymmetric cortical contractility and local cortical extension have been shown to contribute to the partitioning of proteins during the asymmetric divisions of *D. melanogaster* neuroblasts [43, 53, 54]. Therefore, we propose that the establishment in the NSMnb of the dorsal-ventral gradient of CES-1 Snail protein is caused through *pig-1* MELK- and *nmy-2* nonmuscle myosin II-dependent directed flows and/or asymmetric cortical contractility. How this is coordinated with other events that occur just prior to the actual NSMnb division, such as nuclear envelop breakdown, is currently unknown. During certain asymmetric cell divisions, such as the division of the QR neuroblast in *C. elegans* L1 larvae or the divisions of neuroblasts in *D. melanogaster*, asymmetric cortical contractility and extension have been shown to determine cleavage furrow placement and, hence, daughter cell sizes, in a spindle-independent manner [19, 54–56]. Therefore, we furthermore propose that the *pig-1* MELK- and *nmy-2* nonmuscle myosin II-dependent cortical contractility on the ventral side of the NSMnb also determines the position of the cleavage furrow and, hence, the sizes of the NSM and NSMsc. Whether this occurs through a spindle-independent mechanism remains to be determined.

### Roles for CES-1 Snail partitioning and cell size control in life vs death decision

We propose that the partitioning of CES-1 Snail into the NSM contributes to the life vs death decision in the NSM neuroblast lineage. By binding to four conserved Snail binding elements (SBEs) downstream of the *egl-1* BH3-only transcription unit, CES-1 can directly repress *egl-1* transcription [34] (Fig 7B). We propose that the partitioning of CES-1 protein into the NSM results in a concentration of CES-1 in the NSM that is sufficient to bind to the four SBEs and repress *egl-1*. In contrast, the concentration of CES-1 in the NSMsc is not sufficient to repress

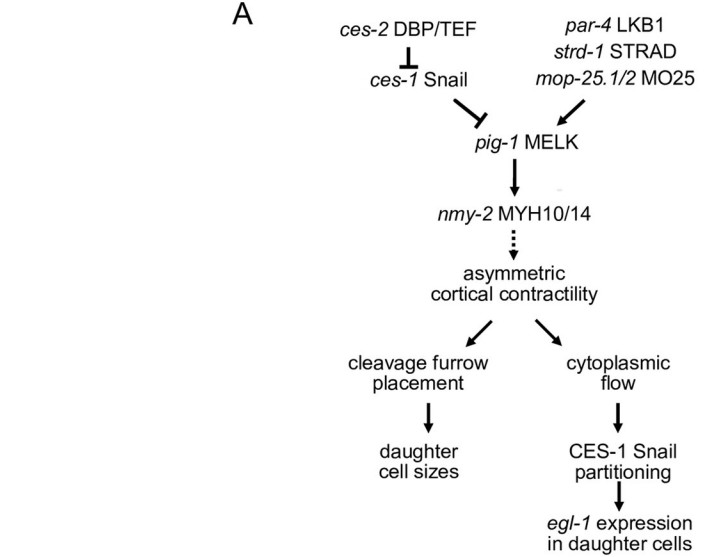

**Fig 7. Coupling of NSM neuroblast division to live vs death decision–Working model. (A)** Genetic pathway that acts in the NSMnb to control the sizes of the daughter cells and the partitioning of CES-1 Snail protein into the NSM. See text for details. **(B)** Schematic representation of cellular events that lead to the asymmetric division by size and fate of the NSMnb. See text for details.

*egl-1*, resulting in the detection of *egl-1* expression 12 min post NSMnb division. We found that in animals lacking *pig-1* MELK function, the NSM and NSMsc are of similar sizes and inherit similar amounts of CES-1 Snail protein. Despite its smaller cell size and reduced amounts of CES-1 protein inherited, the NSM still survives, suggesting that the concentration of CES-1 remains high enough to repress *egl-1* transcription. In the NSMsc, its larger cell size and increased amounts of CES-1 protein lead to a delay in *egl-1* expression and cell death execution by about 10 min. In 98% of the cases, the cell still dies; however, the amount of EGL-1 that has accumulated at the time of death (based on reporter gene expression) is significantly higher compared to wild type, suggesting that cell death execution in these cells is compromised. In animals that lack *ces-2* DBP function, the NSM and NSMsc are of similar sizes [25] and inherit similar amounts of CES-1 Snail protein. However, in *ces-2* mutants, the death of the NSMsc is blocked [24]. Therefore, there is a functional difference between the NSMsc in *pig-1* and *ces-2* mutants with respect to their propensities to die. As shown in **Fig** 5A, the level of CES-1 Snail protein in the NSM neuroblast lineage is increased several-fold in *ces-2* mutants compared to wild type or *pig-1* mutants. This increased level of CES-1 could explain the difference with respect to NSMsc survival between *pig-1* and *ces-2* mutants. However, additional factors that affect *egl-1* transcription and/or cell death execution may also contribute. For example, a heterodimer of the bHLH proteins HLH-2 and HLH-3 (HLH-2/HLH-3) activates *egl-1* transcription in the NSM neuroblast lineage by competing with CES-1 Snail for binding to the SBEs [34] (**Fig** 7B). If the partitioning and/or activity of HLH-2/HLH-3 were affected in either *ces-2* or *pig-1* mutants, a difference in the ability of CES-1 Snail protein to repress *egl-1* transcription in the NSMsc might be observed. Furthermore, a gradient of active CED-3 caspase is formed in the NSM neuroblast and this gradient results in the partitioning of active CED-3 predominantly into the NSMsc, where it promotes cell death execution [36]. The loss of *ces-2*, but not *pig-1*, abolishes the formation of this gradient [36, 38]. Consequently, cell death execution most likely is compromised in the NSMsc in *ces-2*, but not *pig-1* mutants. In conclusion, we consider it likely that factors other than CES-1 Snail contribute to the life vs death decision in the NSM neuroblast lineage, and our results suggest that the activities of these factors may be independent of cell size. However, since we cannot manipulate daughter cell sizes in the NSM neuroblast lineage without affecting levels and/or partitioning of CES-1 Snail protein, we are currently unable to determine their relative contributions.

## A CES-1 Snail auto-regulatory loop

We demonstrate that *ces-1* Snail can control its own activity by inducing the formation of a gradient of CES-1 protein prior to cell division, which then results in the inheritance of different amounts of CES-1 by the two daughter cells. To our knowledge, this is the first time such a mechanism has been described for a member of the Snail family of transcription factors. This auto-regulatory loop depends on the ability of the CES-1 Snail transcription factor to control the expression of the *pig-1* MELK gene, the phosphorylation of nonmuscle myosin II NMY-2 through the kinase PIG-1, and the NMY-2-dependent formation of a gradient of CES-1 protein. The *nmy-2* gene may also be a direct transcriptional target of CES-1 [27], which suggests that CES-1 might control the expression of several components of this regulatory loop. Furthermore, we have previously shown that by controlling the expression of the gene *cdc-25.2* CDC25, CES-1 Snail can slow cell cycle progression in the NSM neuroblast lineage [28], and

this may provide the time necessary for the completion of the regulatory loop, i.e. the formation of the CES-1 gradient.

The auto-regulatory loop of *ces-1* Snail causes its activity to differ in the three different cells of the NSM neuroblast lineage, with the highest activity in the NSM and the lowest in the NSMsc (assuming that the total amount of CES-1 protein in the lineage remains constant). Importantly, this may cause differences in CES-1 target gene selection and, hence, *ces-1* function between these cells. Snail-like transcription factors were originally identified based on their conserved roles in epithelial-mesenchymal transitions (EMTs) [57]. However, more recently, they have also been implicated in various aspects of stem cell biology and function [58, 59]. Interestingly, studies of intestinal stem cell lineages in both *D. melanogaster* and mouse have revealed that the roles of Snail-like transcription factors in stem cells are complex and involve important functions in both stem cell maintenance and daughter cell fate specification. We speculate that auto-regulatory loops similar to the one described here for *C. elegans* CES-1 Snail may contribute to the complex roles of Snail-like transcription factors in stem cell lineages in other organisms.

### Regulation and functions of MELK

We confirm previous reports that in *C. elegans*, the genes *pig-1* MELK and *par-4* LKB1 act in the same genetic pathway [20, 23]. Therefore, *C. elegans* PIG-1 MELK is probably activated through PAR-4 LKB1-dependent phosphorylation. Hence, in the NSM neuroblast lineage, *pig-1* MELK activity is controlled at the level of gene expression through *ces-1* Snail-dependent transcriptional repression [27] and at the post-translational level through *par-4* LKB1-dependent protein phosphorylation.

The role of *C. elegans pig-1* MELK in asymmetric cell division is well established. We present evidence that in this context, PIG-1 MELK kinase acts through the phosphorylation of NYM-2 nonmuscle myosin II. NMY-2 activity then determines the sizes of the resulting daughter cells and the partitioning of one or more factors that contribute to the acquisition of the appropriate fates by the daughter cells. It has been suggested that *C. elegans pig-1* MELK is also required for 'programmed cell elimination' [20, 21], i.e., the extrusion of inappropriately surviving cells from embryos that lack *ced-3* caspase activity. Based on the data presented here, the 'undead' cells present in *pig-1* MELK mutants are not only larger in size but also inherit important factors that they normally would not (e.g., CES-1 Snail, in the case of the NSM neuroblast lineage). We propose that these changes increase the likelihood that inappropriately surviving cells are successfully incorporated into the embryo, rather than being extruded.

Vertebrate MELK may function in a general aspect of cell fate acquisition and its overexpression can promote tumorigenesis [3, 4, 9]; however, the underlying mechanism(s) are not fully understood. We demonstrate that by controlling actomyosin-dependent processes, *C. elegans* PIG-1 MELK affects daughter cell size asymmetry and partitioning of cell fate determinants during asymmetric cell division. This raises the question of whether vertebrate MELK affects cell fate acquisition by similarly controlling actomyosin-dependent processes during asymmetric cell division. Defects in asymmetric cell division have been shown to promote tumorigenesis in a number of cellular contexts [60]. Therefore, overexpression of MELK may promote tumorigenesis by causing defects in asymmetric cell division.

## Methods

### Strains and genetics

Unless noted otherwise, all *C. elegans* strains were maintained at 20˚C as described [61]. Bristol N2 was used as the wild-type strain. Mutations and transgenes used are: LGI: *ces-1*(*tm1036*)

(National BioResource Center) [28], *ces-2*(*bc213*) [25], *nmy-2*(*ne3409*ts) [62], *nmy-2*(*cp13*) (*nmy-2::gfp+LoxP*) [42], *ced-1*(*e1735*) [63]. LGII: *mop-25.2*(*ok2073*)/*mIn1*[*mIs14 dpy-10* (*e128*)] [64], *bcSi66* (P$_{ces-1}$*ces-1::mNeonGreen*) (this study), *ltSi202* (P$_{spd-2}$*gfp::spd-5*) [65]. LGIII: *strd-1*(*ok2283*) [64], *bcIs57* (P$_{pie-1}$*gfp::ph$^{PLCδ}$*) [25], *bcIs66* (P$_{tph-1}$*his-24::gfp*) [28], *bcSi96* (P$_{nmy-2}$*nmy-2::mKate2*) (this study), *bcSi103* (P$_{nmy-2}$*nmy-2$^{S211D/S1974D}$::mKate2*) (this study). LGIV: *pig-1*(*gm344*) [14], *ced-3*(*n2427*) [66], *bcSi102* (P$_{nmy-2}$*nmy-2$^{S211D/S1974D}$::mKate2*) (this study). LGV: *par-4*(*it57*ts) [67], *ltIs44* (P$_{pie-1}$*mCherry::ph$^{PLCδ}$*) [68], *bcIs37* (P$_{egl-1}$*his-24::gfp*) [34], *bcSi97* (P$_{nmy-2}$*nmy-2::mKate2*) (this study), *bcSi101* (P$_{nmy-2}$*nmy-2$^{S211A/S1974A}$::mKate2*) (this study). LGX: *bcSi99* (P$_{nmy-2}$*nmy-2$^{S211A/S1974A}$::mKate2*) (this study), *bcIs65* (P$_{tph-1}$*his-24:: gfp*) (this study).

## Molecular biology

Plasmid pBC1821 was generated using T4 ligation. Briefly, using the primer pairs *ces-1* mNeon-GreenF1 and *ces-1* mNeonGreenR1, *ces-1* mNeonGreenF2 and *ces-1* mNeonGreenR2, and *ces-1* mNeonGreenF3 and *ces-1* mNeonGreenR3, three overlapping DNA fragments were generated and combined using the primer pair *ces-1* mNeonGreenF1 and *ces-1* mNeonGreenR3. The resulting DNA fragment was cloned into pBC1448 [27] using the restriction sites Sph I and Rsr II and T4 ligase to generate plasmid pBC1821 (P$_{ces-1}$*ces-1::mNeonGreen*). Plasmids pBC1876 (P$_{nmy-2}$*nmy-2*), pBC1877 (P$_{nmy-2}$*nmy-2$^{S211AS1974A}$*) and pBC1878 (P$_{nmy-2}$*nmy-2$^{S211DS1974D}$*) were generated from plasmid pAC211 (P$_{nmy-2\ 5.2kb}$*nmy-2*re-encoded::*mKate2::Streptag* 3'UTR$_{nmy-2}$ $_{1.25kb}$ in pCFJ151; a region of ∼ 400 bp in exon 12 of *nmy-2* is re-encoded; *mKate2::Streptag* was added at the 3' of the *nmy-2* ORF; *mKate2* was amplified from a plasmid that we received as a gift from Henrik Bringmann (University of Marburg)). Briefly, using Gibson cloning, plasmid pAC211 was used as template to mutate the codons of the *nmy-2* coding region coding for S211 and S1974 to A211 and A1974 or D211 and D1974 using PCR with high fidelity fusion polymerase. The resulting fragments were digested with SmaI and cloned into pCFJ909 using Gibson ligation to generate pBC1876 (P$_{nmy-2}$*nmy-2*), pBC1877(P$_{nmy-2}$*nmy-2$^{S211A/S1974A}$*) and pBC1878 (P$_{nmy-2}$*nmy-2$^{S211D/S1974D}$*).

## Transgenic animals

Germline transformations were performed as described by Mello and Fire [69]. For the generation of the P$_{ces-1}$*ces-1::mNeonGreen* mosSCI lines, plasmid pBC1821 was injected at a concentration of 40 ng/μl with co-injection markers (pCFJ601 at 50 ng/μl, pGH8 at 10 ng/μl, pCFJ90 at 2.5 ng/μl, pCFJ104 at 5 ng/μl) into the Universal MosSCI strain EG8079 [70] or mosSCI strain EG6699 [71] to acquire the integrated (single-copy) transgenes *bcSi66* (P$_{ces-1}$*ces-1::mNeonGreen*) and *bcSi67* (P$_{ces-1}$*ces-1::mNeonGreen*) on chromosome II. *bcSi66* was chosen for analyses. For the generation of the P$_{nmy-2}$*nmy-2::mKate2* miniMOS line, plasmid pBC1876 was injected at a concentration of 15 ng/μl with co-injection markers (pCFJ601 at 50 ng/μl, pGH8 at 10 ng/μl, pCFJ90 at 2.5 ng/μl, pCFJ104 at 5 ng/μl) into the miniMOS strain HT1593 [70]. Single-copy integrations on chromosome III (P$_{nmy-2}$*nmy-2::mKate2, bcSi96* III) and chromosome V were obtained (P$_{nmy-2}$*nmy-2::mKate2, bcSi97* V). To generate the P$_{nmy-2}$*nmy-2$^{S211AS1974A}$::mKate2* and P$_{nmy-2}$*nmy-2$^{S211DS1974D}$::mKate2* miniMOS lines, plasmids pBC1877 and pBC1878 were injected at a concentration of 10 ng/μl with co-injection markers (pCFJ601 at 50 ng/μl, pGH8 at 10 ng/μl, pCFJ90 at 2.5 ng/μl, pCFJ104 at 5 ng/μl) into the miniMOS strain HT1593 [70] and various single-copy integrations on different chromosomes were obtained (P$_{nmy-2}$*nmy-2$^{S211A/S1974A}$:: mKate2, bcSi99* X, *bcSi101* IV) (P$_{nmy-2}$*nmy-2$^{S211D/S1974D}$::mKate2, bcSi102* IV, *bcSi103* III). For this study, *bcSi97* (P$_{nmy-2}$*nmy-2::mKate2*), *bcSi101* (P$_{nmy-2}$*nmy-2$^{S211A/S1974A}$::mKate2*) and

*bcSi102* (P$_{nmy-2}$*nmy-2*$^{S211D/S1974D}$::*mKate2*) were chosen for analyses. The miniMOS integration sites on different chromosomes were determined using inverse PCR [72].

## Phenotypic analyses

**NSMsc survival.** The number of surviving NSMsc was determined in L3 or L4 larvae using the transgenes *bcIs66* (P$_{tph-1}$*his-24*::*gfp*) or *bcIs65* (P$_{tph-1}$*his-24*::*gfp*) as described [27, 28].

**Quantification of cell volume ratio [NSMsc/NSM].** To estimate the volumes of the NSM and NSMsc, the transgenes P$_{pie-1}$*gfp*::*ph*$^{PLC\delta}$ (*bcIs57*) (*par-4*(*it57*ts) background) or P$_{pie-1}$*mCherry*::*ph*$^{PLC\delta}$ (*ltIs44*) (all other genetic backgrounds) were used to label cell boundaries. The cell volume of NSM and NSMsc was acquired and the cell volume ratio [NSMsc/NSM] determined as described [36]. Imaging was performed using a Leica TCS SP5-II confocal microscope with the laser power set to 13%.

**Kinetics of NSMsc cell death.** 4D Nomarski microscopy was used to determine the time it takes individual NSMsc to die [73]. Briefly, recordings were started prior to NSMnb division and the NSMsc was tracked until it formed a refractile cell corpse essentially as described [27].

**Quantification of P$_{egl-1}$*his-24*::*gfp* expression.** The kinetics of *egl-1* transcription in the NSMsc was determined using the transcriptional reporter P$_{egl-1}$*his-24*::*gfp* (*bcIs37*). Transgene P$_{pie-1}$*mCherry*::*ph*$^{PLC\delta}$ (*ltIs44*) was used to visualize the cell boundary of the NSMsc. The fluorescence intensity of P$_{egl-1}$*his-24*::*gfp* in the NSMsc was measured in the center Z-slice of a Z-stack of the NSMsc using Image J and divided by the area of that slice. Values for fluorescence intensity/area were normalized to the values at 0 min, which were set to 1. Imaging was performed using a Leica TCS SP5-II confocal microscope and the laser power for detecting P$_{pie-1}$*mCherry*::*ph*$^{PLC\delta}$ and P$_{egl-1}$*his-24*::*gfp* were set to 13% or 30%, respectively.

**Quantification of P$_{ces-1}$*ces-1*::mNeonGreen.** The single-copy transgene P$_{ces-1}$*ces-1*::*mNeonGreen* (*bcSi66*) was used to visualize CES-1 protein and the transgene P$_{pie-1}$*mCherry*::*ph*$^{PLC\delta}$ (*ltIs44*) was used to label cell boundaries. To detect CES-1::mNeonGreen in the NSMnb lineage, a laser power of 60% had to be used (Leica TCS SP5-II confocal microscope). Because of bleaching, recordings were done at three different time points (~5 min before NSMnb metaphase, at NSMnb metaphase and post NSMnb division). Maximum intensity projections of Z-stacks through the NSMnb, NSM or NSMsc were generated and used for analyses. Total fluorescence intensity/area at different points along the dorsal to ventral axis of the NSMnb (0–4.8 μM) was determined in overlapping 2 pixel-steps. Values obtained for a certain genotype were normalized to the value obtained for the most dorsal area (0.0 μM), which was set to 1. To determine the ratios of total fluorescence intensities in the ventral to dorsal half of the NSMnb, the NSMnb in the maximum intensity projection was divided into dorsal and ventral halves, the fluorescence intensities in the two halves measured, and the two values obtained divided [ventral/dorsal]. The same procedure was used to determine the ratios of total fluorescence intensities in the NSM to NSMsc [NSM/NSMsc].

**Visualization and quantification of *nmy-2*::*gfp*+*LoxP*(*cp13*).** The CRISPR knock-in allele *nmy-2*(*cp13*) (*nmy-2*::*gfp*+*LoxP*) was used to visualize endogenous NMY-2 protein (green). Cell boundaries were labeled with transgene P$_{pie-1}$*mCherry*::*ph*$^{PLC\delta}$ (*ltIs44*) (magenta). For image acquisition, 25% laser power (P$_{nmy-2}$*nmy-2*::*gfp*) or 13% laser power (P$_{pie-1}$*mCherry*::*ph*$^{PLC\delta}$) was used (Leica TCS SP5-II). All analyses were performed on center Z-slices of Z-stacks through the NSMnb. For determining fluorescence intensity/area at different points along the dorsal to ventral axis of the NSMnb (0–4.8 μM), we measured fluorescence intensity/area in overlapping 2 pixel-steps. Values obtained for a certain genotype were normalized to the value obtained at 2.4 μM, which was set to 1. To determine the ratios of fluorescence intensities in the ventral to dorsal half of the NSMnb, the NSMnb in the center Z-slice was divided

into dorsal and ventral halves, the fluorescence intensity in the two halves were measured and the values obtained divided [ventral/dorsal].

**Determination of the orientation of the NSMnb cleavage plane.** The orientation of the NSMnb cleavage plane was analyzed using transgene $P_{pie-1}mCherry::ph^{PLC\delta}(ltIs44)$ (magenta), which labels cell boundaries, and transgene $P_{spd-2}gfp::spd-5(ltSi202)$, which labels centrosomes (green), as described previously [27].

**Quantification of embryonic lethality.** Briefly, ~40–50 young adults of each genotype were dissected to acquire early-stage embryos. These embryos were transferred to slides with 2% agarose pads, cover slips were added and sealed with petroleum jelly and slides incubated at 25°C for ~7 h after which the viability of embryos was determined using light microscopy.

## Confocal microscopy

Confocal imaging was done using a Leica TCS SP5-II confocal microscope. Different fusion proteins required different laser powers for imaging but for each fusion, the laser power was kept constant throughout the experiments. Strains were maintained at 15°C (temperature-sensitive alleles) or 20°C. Three hours prior to confocal recording, ~10 to 15 adults were dissected to acquire early-stage embryos. These embryos were transferred to slides with 2% agarose pads, cover slips were added and sealed with petroleum jelly, and slides incubated at 25°C. For recording of the NSMnb lineage, a Z-stack of 7.5–8.0 μm with a step distance of 0.5 μm was acquired.

## RNA interference

*mop-25.1* and *mop-25.2* RNAi was done using the microinjection method as described [74]. *mop-25.1* and *mop-25.2* dsRNAs were both injected at a concentration of 100 ng/μl into *mop-25.2*(*ok2073*)/*mIn1*[*mIs14 dpy-10*(*e128*)] young adults. Injected adults were maintained at 25°C for ~22–24 h before imaging F1 embryos. For quantifying NSMsc survival of *mop-25.2* (*ok2073*)/*mIn1*[*mIs14 dpy-10*(*e128*)], injected *mop-25.2*(*ok2073*)/*mIn1*[*mIs14 dpy-10*(*e128*)] adults were removed after a ~24 h incubation period at 25°C and NSMsc survival was determined in F1 progeny once they had reached the L3 or L4 stage.

## Handling of temperature-sensitive (ts) mutants

To image the NSMnb lineage in *par-4*(*it57*ts) animals, L4 larvae were picked, maintained at 25°C for 16 h and then transferred to 15°C overnight [23]. Early stage embryos were acquired by dissecting the resulting adults, incubating the embryos at 20°C for 1 h and subsequently at 25°C for 1 h until they had reached the early comma stage. For determining NSMsc survival in the *par-4*(*it57*ts) background, L4 larvae were incubated at 25°C for ~20 h and shifted to 20°C for another ~16 h to let them lay F1 embryos. The P0 adults were removed and NSMsc survival determined in F1 animals, once they had reached the L3 or L4 stage at 15°C. To image the NSMnb lineage in *nmy-2*(*ne3409*ts) animals, adults were maintained at 15°C, early stage F1 embryos were acquired by dissecting P0 adults, incubated at 15°C for 1 h, at 20°C for 45 min, and at 25°C for ~15 min before confocal imaging. For determining NSMsc survival rate in *nmy-2*(*ne3409*ts) animals, L4 larvae were picked and maintained at 15°C overnight. The following day, the young P0 adults were shifted to 25°C for 10 min and transferred back to 15°C for 40 min. These temperature shifts were repeated for ~9 to 10 h after which the P0 animals were removed. The F1 embryos were allowed to develop to the L3 or L4 stage at 15°C and subsequently analyzed.

## Supporting information

**S1 Fig. Daughter cell sizes in the NSM neuroblast lineage. (Left)** Series (Z-stacks from top to bottom) of fluorescence confocal images of the NSM and NSMsc immediately after NSMnb division in wild-type and mutant embryos expressing the transgene $P_{pie-1}gfp::ph^{PLC\delta}$ (*bcIs57*) (*par-4*(*it57*ts) mutant background) or $P_{pie-1}mCherry::ph^{PLC\delta}$ (*ltIs44*) (all the other mutant backgrounds), which expresses a fusion protein that labels the cell boundary (magenta). Grey and white arrow heads point to the NSMsc or NSM, respectively. **(Right)** Schematic representations of the areas of the NSM (white) or NSMsc (grey) for each genotype and corresponding cell volume ratios [NSMsc/NSM]. Scale bar 5 μm.
(TIF)

**S2 Fig. Kinetics of the NSM sister cell death.** Nomarski images of representative wild-type and mutant embryos at different times post fertilization [min]. 410 min is immediately post NSMnb cytokinesis. Insets show the NSMsc and white arrow heads point to relevant cells. Scale bar 10 μm.
(TIF)

**S3 Fig. Kinetics of upregulation of *egl-1* expression in the NSM sister cell.** Fluorescence images of NSMsc in representative wild-type [+/+] and *pig-1*(*gm344*) embryo carrying the transgenes $P_{egl-1}his-24::gfp$ (*bcIs37*) (green) at different time points post NSMnb division [min]. Cell boundaries are labeled with transgene $P_{pie-1}mCherry::ph^{PLC\delta}$ (*ltIs44*) (magenta). White arrow heads point to NSMsc. Scale bar 2 μm.
(TIF)

**S4 Fig. $P_{ces-1}ces-1$::*mNeonGreen* rescues polarity defect of *ces-1* Snail loss-of-function (lf) mutant. (A)** Series (Z-stacks from top to bottom) of fluorescence confocal images of three different *ces-1*(*tm1036*) embryos (0˚, 90˚, -45˚) at NSMnb metaphase (M) and in NSMnb division (ID). All embryos were homozygous for transgene $P_{spd-2}::gfp::spd-5$(*ltSi202*) (green), which labels the centrosomes, and for transgene $P_{pie-1}mCherry::ph^{PLC\delta}$ (*ltIs44*) (magenta), which labels cell boundaries. Grey arrow heads point to the centrosomes, which are segregated to the NSMsc, and white arrow heads point to the centrosomes, which are segregated into the NSM post NSMnb division. 0˚ indicates the 'wild-type' orientation of the NSMnb cleavage plane. 90˚ and -45˚ indicate 'mutant' orientations of the NSMnb cleavage plane found in *ces-1*(*tm1036*) mutants. **(B)** Quantification of the fraction of different orientations of the NSMnb cleavage plane in embryos of various genotypes (n = 16–17). Transgene $P_{ces-1}ces-1$::*mNeonGreen* (*bcSi66*) fully rescues the defect in the orientation of the NSMnb cleavage plane observed in *ces-1*(*tm1036*).
(TIF)

**S5 Fig. Kinetics of asymmetric enrichment of nonmuscle myosin NMY-2 in the NSM neuroblast. (A)** Series of fluorescence confocal images of *nmy-2*::*gfp*+*LoxP* (*cp13*) (green) in representative NSMnb in wild-type [+/+] and *pig-1*(*gm344*) embryo. Recordings were performed from ~ -11 min prior to NSMnb division until the NSMnb divided (0 min). Cell boundaries are labeled with transgene $P_{pie-1}mCherry::ph^{PLC\delta}$ (*ltIs44*) (magenta). D indicates dorsal and V ventral. White dotted line indicates the middle of the NSMnb. Scale bar 2 μm. **(B)** Shown are mean ratios of fluorescence intensities + SEM [$P_{nmy-2}nmy-2$::*gfp*] in the ventral to dorsal half of the NSMnb at different times prior to NSMnb division are shown (n = 3). Vertical **red** dotted indicates the time point at which asymmetry of NMY-2::GFP in wild type was established. For each genotype, the mean ratios obtained were normalized to the mean ratio at recording time -11.2 min, which was set to 1. A ratio of 1.0 is indicated by the horizontal black dotted line.
(TIF)

**S6 Fig. Total amount of endogenous NMY-2::GFP. (A)** Fluorescence confocal images of *nmy-2*::*gfp*+*LoxP* (*cp13*) in representative wild-type [+/+] and *pig-1*(*gm344*) embryo ~5 min before the NSMnb metaphase and at NSMnb metaphase. Insets show the NSMnb and white arrow heads point to the NSMnb, D and V indicate dorsal and ventral half of the NSMnb. White dotted lines indicate middle of NSMnb. Scale bar 10 μm. **(B)** Quantification of NMY-2:: GFP in embryos. Shown are mean total fluorescence intensities per embryo in wild type [+/+] and *pig-1*(*gm344*) ~5 min before NSMnb metaphase and at NSMnb metaphase (n = 12–14). All embryos analyzed were homozygous for the transgene *nmy-2*::*gfp*+*LoxP*(*cp13*) (green). Statistical significance was determined using Mann–Whitney test (ns, no significance). **(C)** Quantification of NMY-2::GFP in NSMnb. Shown are mean total fluorescence intensities per NSMnb in wild type [+/+] and *pig-1*(*gm344*) ~5 min before NSMnb metaphase, at NSMnb metaphase and at the time the cleavage furrow forms (n = 12–14). All embryos analyzed were homozygous for the transgene *nmy-2*::*gfp*+*LoxP*(*cp13*) (green). Statistical significance was determined using Mann–Whitney test (ns, no significance).
(TIF)

**S7 Fig. Phosphorylation at residues S211 and S1974 is critical for the ability of nonmuscle myosin NMY-2 to promote embryonic viability.** Embryonic viability [%] at 25°C in different genotypes. *nmy-2*(S), *nmy-2*(S-D), *nmy-2*(S-A) represent single-copy transgenes $P_{nmy-2}nmy-2$ (*bcSi97*), $P_{nmy-2}nmy-2^{S211DS1974D}$ (*bcSi102*) and $P_{nmy-2}nmy-2^{S211AS1974A}$ (*bcSi101*), respectively. For each genotype, three independent experiments were performed (n = 240–290). Date are presented as mean ± SEM.
(TIF)

**S8 Fig. Phosphorylation at residues S211 and S1974 is critical for the ability of nonmuscle myosin NMY-2 to control daughter cell size in the NSM neuroblast lineage. (A)** Schematic representations of wild-type and mutant NMY-2 proteins. Mutant protein NMY-2(S-A) cannot be phosphorylated at S211 and S1974. Mutant protein NMY-2(S-D) mimics wild-type NMY-2 protein phosphorylated at S211 and S1974. Wild-type NMY-2 protein is referred to as NMY-2(S) **(B)** Cell volume ratio [NSMsc/NSM] in different genotypes. All strains were homozygous for the transgene $P_{pie-1}mCherry::ph^{PLC\delta}$ (*ltIs44*). *nmy-2*(S) 2nd, *nmy-2*(S-D) 2nd and *nmy-2*(S-A) 2nd represent the integrated single-copy transgenes $P_{nmy-2}nmy-2$ (*bcSi96*), $P_{nmy-2}nmy-2^{S211DS1974D}$ (*bcSi103*) and $P_{nmy-2}nmy-2^{S211AS1974A}$ (*bcSi99*), respectively. Each grey dot represents the ratio of one pair of daughter cells (n = 9–11). Red horizontal lines represent the mean ratio obtained for a given genotype, which is also shown on top. The black dotted horizontal line at a ratio of 0.64 represents the ratio of wild type. Statistical significance was determined using Mann–Whitney test (****, P≤0.0001, ns, no significance).
(TIF)

**S1 Data. Numerical data presented in figures.**
(XLSX)

## Acknowledgments

The authors thank M. Amoyel, N. Goehring, R. Poole, T. Mullan and A. Sethi for comments on the manuscript and members of the Conradt and Zanin laboratories for discussion; M. Bauer, N. Lebedeva, and M. Schwarz for excellent technical support; Y.W. Jiang for help with genotyping; H. Bringmann for providing the *nmy-2*::*mKate2* template. Some strains used in this study were provided by the Caenorhabditis Genetics Center (CGC; https://cbs.umn.edu/cgc/home) and the National BioResource Project (NBRP; https://shigen.nig.ac.jp/c.elegans/).

## Author Contributions

**Conceptualization:** Hai Wei, Eric J. Lambie, Barbara Conradt.

**Data curation:** Hai Wei.

**Formal analysis:** Hai Wei, Barbara Conradt.

**Funding acquisition:** Hai Wei, Ana X. Carvalho, Barbara Conradt.

**Investigation:** Hai Wei, Eric J. Lambie, Daniel S. Osório.

**Methodology:** Hai Wei, Eric J. Lambie.

**Project administration:** Barbara Conradt.

**Resources:** Daniel S. Osório, Ana X. Carvalho.

**Supervision:** Eric J. Lambie, Ana X. Carvalho, Barbara Conradt.

**Validation:** Hai Wei, Eric J. Lambie, Barbara Conradt.

**Visualization:** Hai Wei.

**Writing – original draft:** Hai Wei, Barbara Conradt.

**Writing – review & editing:** Hai Wei, Eric J. Lambie, Daniel S. Osório, Ana X. Carvalho, Barbara Conradt.

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
