## [Decision Letter · Decision Letter 0]

30 Jun 2020

Dear Dr. Conradt

Thank you very much for submitting your Research Article entitled 'PIG-1 MELK-dependent phosphorylation of nonmuscle myosin II promotes apoptosis through CES-1 Snail partitioning' to PLOS Genetics. Your manuscript was fully evaluated at the editorial level and by three independent peer reviewers. The reviewers appreciated the attention to an important topic, but requested a number of minor edits.

We therefore ask you to modify the manuscript according to the review recommendations before we can consider your manuscript for acceptance. Your revisions should address the specific points made by each reviewer.

Accompanying reviewer attachments should be included with this email; please notify the journal office if any appear to be missing. They will also be available for download from the link below. You can use this link to log into the system when you are ready to submit a revised version, having first consulted our Submission Checklist.

[LINK]

Best wishes,

Shai Shaham

Guest Editor

PLOS Genetics

Gregory P. Copenhaver

Editor-in-Chief

PLOS Genetics

Reviewer's Responses to Questions

**Comments to the Authors:**

Reviewer #1: See attached document.

Reviewer #2: Summary

Asymmetric partitioning of cellular substances is an important mechanism for establishing cell asymmetry and determining cell fate. The C. elegans protein kinase PIG-1 is known to regulate asymmetrical cell division. This report further reveals the molecular mechanism PIG-1 utilizes to regulate downstream events, focusing on the regulation of apoptosis of the NSM sister cell. The authors found that CES-1, a critical repressor of the BH3-domain only protein and cell death inducer EGL-1, is asymmetrically distributed into the two daughter cells of NSMnb, one of which becomes NSM, the other undergoes apoptosis. Based on the genetic and cell biological study results reported here and on the previously published information about PIG-1, the authors propose a model that PIG-1 phosphorylates the non-muscle myosin molecule NMY-2 and this phosphorylation results in the asymmetric distribution of NMY-2 to the ventral side of the NSMnb. This event further facilitates the asymmetric division of the NSMnb into two daughter cells with different sizes, and the asymmetric distribution of CES-1; as a consequence, EGL-1 expressed is less repressed in the smaller sister cell and leads to the apoptosis of the smaller sister cell, whereas the larger sister cell becomes NSM. Furthermore, the authors showed that the C. elegans par-4, strd-1 or mop-25.1, 2 genes, which encode C. elegans homologs of LKB1, STRADα and MO25α, acts in the same pathway as pig-1 does, and propose that these genes act upstream of PIG-1 to activate it. In addition, integrating the authors’ previous finding, here the authors also propose an autoregulatory loop of CES-1 in which CES-1 inhibits the activity of PIG-1, which in turn regulates the asymmetric distribution of CES-1.

Significance

Although almost all apoptosis events are executed through a caspase-dependent pathway, there are many different upstream pathways that activate caspases in particular cells, in response to different signals. This report has discovered a specific apoptosis-initiation mechanism that acts through asymmetric distribution of a snail family transcription repressor that targets the expression of the cell death initiator EGL-1. This pathway is led by protein kinase PIG-1 and involves non-muscle myosin and thus probably the actomyosin network in generating a dorsal-ventral gradient of certain proteins inside a mother cell during cell division. This novel mechanism of apoptosis-initiation not only reveals the previously unknown roles of PIG-1 and the PAR-4 pathway proteins, but also broadens our knowledge regarding how many different ways apoptosis can be initiated.

Mammalian Maternal Embryonic Leucine zipper Kinase (MELK) protein is known to promote tumorigenesis, however, the mechanism behind this function remains unknown. On the other hand, C. elegans PIG-1, the ortholog of MELK, was reported to regulate asymmetrical cell fate in multiple lineages. The finding reported here will definitely shed light on our understanding of the function and regulation of mammalian MELK, and further on the mechanisms of tumorigenesis.

The auto-regulatory loop of CES-1 contributes to the precise control of CES-1 protein level in the three types of cells: NSMnb, NSM, and NSM sister cells. The discovery of this loop suggests that other snail family proteins might use the same regulatory mechanisms in cells.

This report thus will be of general interest to a broad audience in the research fields of cell biology, developmental biology, and cancer biology.

The experiments reported here are properly performed with the right controls and the results are rigorously analyzed, with repeats and quantifications. The conclusions are drawn based on the results and thus are solid. The models proposed here have integrated the findings made in this report and in the literature and thus are convincing. The report is clearly written. I enjoy reading this manuscript very much.

Specific points

A few points to be addressed are listed here. I do not intend to request new experiments. Addressing points 1 and 2 in Discussion, clarifying points 3 in the text, and revising point 4 would be sufficient.

1. The subcellular localization of CES-1::mNG (mNeonGreen).

Is CES-1 a nuclear protein? According to Fig. 5A, CES-1::mNG is in the cytoplasm or at least all over the cell even prior to metaphase. Is the asymmetrical partitioning of CES-1 coordinated with the nuclear envelope breakdown during mitosis? When does nuclear envelope breakdown occur in the context of CES-1 partitioning, which appears a rapidly dynamic event?

2. As the protein kinase that phosphorylates NMY-2, is PIG-1 also localized to the cell cortex and enriched on the ventral side in NSMnb like NMY-2? Knowing this information would potentially strengthen one part of the model proposed, that is, NMY-2 is a direct substrate of PIG-1.

3. Fig. S4A was not mentioned in the text.

4. Fig. S5A: NMY-2::GFP time-lapse images are over-exposed, making it hard to judge the signal intensity.

Reviewer #3: MELK is over-expressed in some cancers, and shown to be required for clonogenic growth of breast cancer cells, although little was known about how it functioned. Pig-1, the C. elegans MELK ortholog, had previously been shown to regulate asymmetric cell division and apoptosis, providing an excellent model system to investigate its function further. This paper provides a thorough analysis of pig-1 in the NSM neuroblast, demonstrating a detailed pathway that may provide insight into MELK’s role in cancer. Overall, the findings are clearly presented, and novel tools are developed that provide new insight. However, many of these findings had previously been shown in other cell types, which reduces some of the novelty. For example, pig-1 had previously been shown to act in the same pathway as par-4, pig-1 had already been shown to affect non-muscle myosin localization, and pig-1 had already been shown to be a target of ces-1. Adjustments to the writing throughout the paper would help emphasize the new findings and indicate which results were consistent with previous findings. The new findings are that pig-1 leads to asymmetric localization of Ces-1 and thereby egl-1, and these are dependent on phosphorylation state of NMY-2. These are important findings that could explain resistance to apoptosis and cell fate changes observed in cancer cells.

Specific suggestions:

1. Fig. 2A and legend. It should be indicated that this is an infrequent event in pig-1 mutants. For example, 2% could be written on the side of the figure. As it is presented, it looks like all of the ‘NSM’ cells live longer.

2. What is the nature of the gm344 allele? Have other alleles been shown to behave similarly? It seems surprising that only one allele is investigated.

3. Fig. 4. It is noticeable that the level of egl-1 also becomes much higher in the pig-1 mutant, suggesting it takes more egl-1 to kill. This could be very interesting and could be mentioned.

4. Egl-1 should be indicated in Fig. 7a.

**Have all data underlying the figures and results presented in the manuscript been provided?**

Reviewer #1: Yes

Reviewer #2: Yes

Reviewer #3: Yes

PLOS authors have the option to publish the peer review history of their article (what does this mean?). If published, this will include your full peer review and any attached files.

Reviewer #1: **Yes: **Brent Derry

Reviewer #2: No

Reviewer #3: No

---

## [Editor Report · Decision Letter 1]

29 Jul 2020

Dear Dr Conradt

We are pleased to inform you that your manuscript entitled "PIG-1 MELK-dependent phosphorylation of nonmuscle myosin II promotes apoptosis through CES-1 Snail partitioning" has been editorially accepted for publication in PLOS Genetics. Congratulations!

Yours sincerely,

Shai Shaham

Guest Editor

PLOS Genetics

Gregory P. Copenhaver

Editor-in-Chief

PLOS Genetics

Comments from the reviewers (if applicable):

**Data Deposition**

http://datadryad.org/submit?journalID=pgenetics&manu=PGENETICS-D-20-00874R1

**Press Queries**

---

## [Editor Report · Acceptance letter]

8 Sep 2020

PGENETICS-D-20-00874R1 

PIG-1 MELK-dependent phosphorylation of nonmuscle myosin II promotes apoptosis through CES-1 Snail partitioning 

Dear Dr Conradt, 

We are pleased to inform you that your manuscript entitled "PIG-1 MELK-dependent phosphorylation of nonmuscle myosin II promotes apoptosis through CES-1 Snail partitioning" has been formally accepted for publication in PLOS Genetics! Your manuscript is now with our production department and you will be notified of the publication date in due course.

With kind regards,

Matt Lyles

PLOS Genetics

On behalf of:
